# Establishment and Hemodynamic Assessment of the Superior Cavopulmonary Anastomosis in a Reproducible Porcine Model

**DOI:** 10.3390/biomedicines13040918

**Published:** 2025-04-09

**Authors:** Benjamin Bierbach, Luca Pieterek, Jan Dauvergne, Carolin Scholl, Christina Oetzmann von Sochaczewski, Johannes Breuer, Boulos Asfour, Mathieu Vergnat, Tobias Kratz

**Affiliations:** 1Department of Pediatric Cardiac Surgery, University Hospital Bonn, 53127 Bonn, Germany; pieterek.luca@gmail.com (L.P.); caro.scholl@online.de (C.S.); boulos.asfour@ukbonn.de (B.A.); mathieu.vergnat@ukbonn.de (M.V.); 2Department of Pediatric Cardiology, University Hospital Bonn, 53127 Bonn, Germany; jan.dauvergne@t-online.de (J.D.); johannes.breuer@ukbonn.de (J.B.); tobias.kratz@ukbonn.de (T.K.); 3Section of Pediatric Surgery, University Hospital Bonn, 53127 Bonn, Germany; christina.oetzmann@ukbonn.de

**Keywords:** Glenn, porcine model, univentricular circulation, cavopulmonary connection, bidirectional cavopulmonary connection, right ventricular failure

## Abstract

**Background**: Palliative surgery for the treatment of functionally univentricular heart malformations consists of a staged approach to separation of the pulmonary and systemic circulation, including the creation of a superior cavopulmonary connection. Literature on the superior cavopulmonary connection in porcine models lacks information on details of the procedure as well as data on its acute hemodynamic effects. In preliminary experiments, we were unable to reproduce an already published porcine model. Therefore, we used a conduit extension and cardiopulmonary bypass in order to achieve hemodynamic stability and still employ the commonly used straight downward pathway for the superior caval vein onto the right pulmonary artery, as in the human clinical setting. This model of a univentricular circulation utilising the superior cavopulmonary anastomosis is intended to be applied in the setting of unilateral diaphragmatic palsy. Hence, we aim to investigate the effect of unilateral diaphragmatic pacing in a reproducible model of univentricular physiology. **Methods**: Therefore, we constructed an anastomosis between the superior caval vein and the right pulmonary artery (RPA) in 14 pigs on cardiopulmonary bypass using a 12 mm expanded polytetrafluorethylene interposition graft. Six pigs received a bidirectional cavopulmonary connection with unrestricted atrial septal communication (BDCPC), while eight pigs received a unidirectional cavopulmonary connection (UDCPC) to the excluded RPA. **Results**: The BDCPC resulted in an impaired cardiopulmonary state (cardiac output dropped from 3.15 ± 0.21 to 2.17 ± 0.19 L/min; *p* < 0.01), mean arterial pressure plummeted (from 80.8 ± 3.7 to 49.3 ± 7.3 mmHg; *p* = 0.02), arterial lactate concentration rose (from 0.82 ± 0.09 to 4.36 ± 0.96 mmol/L; *p* = 0.01), arterial oxygen saturation dropped (from 95.8 ± 1.1 to 60.9 ± 10.4%; *p* < 0.01), and right ventricular function deteriorated (tricuspid annular plane systolic excursion decreased from 12 ± 0.7 to 5 ± 0.7 mm; *p* < 0.01). In contrast, in the UDCPC group, the cardiopulmonary parameters indicated a stable condition. **Conclusions:** Consequently, a UDCPC is a more suitable acute model in pigs for a univentricular circulation. The model’s reproducibility may aid in future research on partial cavopulmonary connection.

## 1. Introduction

The treatment of patients with a functionally univentricular heart consists of a staged circulatory separation [1]. Systemic perfusion is maintained by the univentricular heart, and pulmonary perfusion is achieved by routing the caval blood flow directly into the pulmonary artery vasculature without a supportive ventricle. The circulatory separation is achieved in a stepwise manner due to the pulmonary arteries’ initial inability to accommodate the required blood volume because of their elevated vascular resistance. During pulmonary artery vasculature maturation, the resistance plummets and, in the course of the first year’s second half, reaches a level at which passive pulmonary blood flow may be achieved [2,3]. Therefore, after initially balancing systemic and pulmonary circulation during the newborn and infant period, the first step in diverting the caval blood flow into the pulmonary circulation consists of creating a superior cavopulmonary anastomosis. At this stage, the SVC’s blood volume may be the only source of pulmonary blood flow, or, in some cases, additional antegrade pulmonary blood flow via a connection to the heart may be present. In any case, this univentricular physiology is characterised by an altered blood flow pattern in the pulmonary artery compared to the normal pulsatile blood flow [4,5]. The loss of pulsatility renders the amount of pulmonary blood flow critically dependent on an operational diaphragm [6]. Unilateral phrenic nerve palsy is one of many long-term sequelae of congenital cardiac surgery [7,8,9,10,11]. Diaphragmatic palsy narrows the amplitude of the intrathoracic pressure gradient typical of physiological respiration, leading to decreased pulmonary blood flow. Therefore, it is of utmost importance to restore diaphragmatic function in the form of a coordinated diaphragmatic inspiratory movement compared to merely plicating the paralysed hemidiaphragm and thus immobilising it for a lifetime [8]. This potential complication is particularly relevant in patients with univentricular heart physiology [12].

In order to effectively study the potential treatment option of a synchronised unilateral diaphragmatic pacemaker, we first intended to establish a model that mimics the cardiopulmonary status of a univentricular heart patient with non-pulsatile pulmonary artery blood flow. Currently, various porcine models for cavopulmonary connections exist [13,14,15,16,17]. In initial experiments, we applied the methodology published by Henaine and coworkers, creating a superior cavopulmonary connection to the RPA on the beating heart without circulatory support [14]. Unfortunately, in our hands, we were unable to replicate the model due to unfavourable porcine anatomy. Our goal of creating a reproducible and cardiopulmonally stable animal model leading to a pulmonary blood flow pattern typical of the univentricular heart was not achievable using this animal model. Therefore, we utilised CPB and compared a bidirectional cavopulmonary connection facilitated by an ePTFE conduit extension plus an unrestrictive atrial communication and central pulmonary artery banding with a unidirectional cavopulmonary connection utilising an ePTFE conduit extension.

## 2. Materials and Methods

The study protocol was approved by the ethics committee for laboratory animals, North Rhine-Westphalia. All animals in this study received humane care in compliance with the *Principles of Laboratory Animals* formulated by the National Society for Medical Research and the *Guide for the Care and Use of Laboratory Animal Resource* published by the National Institutes of Health, Bethesda, Md. This study was also conducted in accordance with the 3Rs (Replacement, Reduction, and Refinement) principles to ensure ethical and responsible animal research. Animals were allocated to the experimental group by lot. Confounders were not controlled. No blinding was performed.

### 2.1. Experimental Groups

In 14 healthy, juvenile German landrace pigs (10 males and 4 females), aged 3 months and weighing 18 to 25 kg, a superior cavopulmonary connection was constructed. The pigs, which had no genetic modifications, were supplied by the Agricultural School of the Rhenish Friedrich Wilhelm University of Bonn, Germany. The animals had a conventional microbiological status and underwent a three-day acclimatisation period at the research facility. The environmental conditions were meticulously regulated, with a temperature maintained between 16 and 18 °C and a relative humidity level of 50–60%. The air was exchanged at least eight times per hour. Each pig was housed individually in a 4–6 m^2^ enclosure, which was enriched with chains, balls, and other play materials. An infrared heating lamp was available at all times to provide warmth during rest periods. A 12 h light–dark cycle was implemented, with artificial lighting provided from 7:00 a.m. to 7:00 p.m. Water was provided ad libitum and the pigs received a standard diet (Altromin 9023, Altromin Spezialfutter, Lage, Germany).

The bidirectional cavopulmonary connection (BDCPC) group (*n* = 6) received an end-to-side anastomosis between the superior vena cava and the right pulmonary artery, an unrestrictive atrial communication, and a tight main pulmonary artery banding. This led to a bidirectional cavopulmonary anastomosis with limited antegrade pulmonary blood flow. In the unidirectional cavopulmonary connection (UDCPC) group (*n* = 8), an end-to-side anastomosis between the SVC and the RPA was created, and the RPA was occluded at its origin. This resulted in a unidirectional cavopulmonary anastomosis. All animals were studied with regard to survival and the level of hemodynamic stability using a standardised respiratory and inotropic support (full ventilation using 20 mbar peak inspiratory pressure (PIP), 10 mbar positive end expiratory pressure (PEEP), 15 breaths per minute, and the heart was weaned off CPB using adrenaline 0.05 µg/kg × min).

### 2.2. Surgical Preparation

Animals were premedicated with 20 mg/kg ketamine (VetViva Richter GmbH, Wels, Austria), 2 mg/kg azaperone (Streuli Pharma AG, Uznach, Switzerland), and 0.02 mg/kg atropine (B. Braun Melsungen AG, Melsungen, Germany). General anaesthesia was induced with 0.5 to 0.6 mg/kg propofol (Fresenius Kabi, Bad Homburg, Germany) and 7.5 mg piritramide (Hameln pharma GmbH, Hameln, Germany). After oral intubation, the animals were mechanically ventilated by a Servo-i respirator (Maquet, Rastatt, Germany) with a respiratory rate of 12 to 15 breaths per minute, aiming for an arterial partial pressure of Carbon Dioxide (pCO_2_) between 35 and 45 mmHg. During the dissection, the baseline measurements and CPB the fraction of inspired oxygen was set to 21%. For weaning off CPB and the remaining one-hour observation period, the fraction of inspired oxygen was set to at least 40%. The lowest accepted arterial oxygen saturation after separation from CPB was 70%. The fraction of inspired oxygen was set accordingly. The respiratory mode was set to synchronised intermittent mechanical ventilation with a peak pressure of 18 to 21 mbar, and a positive end expiratory pressure of 5 mbar was applied. Anaesthesia was maintained intravenously with 0.3 to 1 mg/kg per hour of propofol and 3 to 5 mg per hour of piritramide. Arterial blood gases were maintained within physiologic ranges (Rapid Point 500, Siemens Healthineers, Forchheim, Germany). An isotonic, iso-oncotic crystalloid solution (Fresenius Kabi, Bad Homburg, Germany) was administered with an hourly rate of 5 to 10 mL/kg. Both femoral arteries were cannulated (20 Gauge Arrow, Teleflex Medical, Athlone, Ireland). A triple lumen central venous catheter (5.5 French (Fr.) Arrow, Teleflex Medical, Athlone, Ireland) was placed in the IVC via the femoral vein.

### 2.3. Surgical Set-Up

The surgical preparation was carried out by an experienced paediatric cardiac surgical team, including a well-versed paediatric perfusionist. The operating surgeon had several years of experience in experimental cardiac research, specifically in pigs. In all animals, a median sternotomy was performed using an oscillating saw (Aesculap, Tuttlingen, Germany), and a sternal retractor (KLS Martin, Tuttlingen, Germany) kept the chest open throughout the entire experiment. The thymus was carefully removed, ensuring preservation of the phrenic nerves and pleura. Both Vv. thoracicae internae were circumferentially dissected from the chest wall for 5 to 6 cm, allowing mobilisation of the caval vein. Additionally, both internal jugular veins and subclavian veins were dissected for further mobilisation of the SVC (Figure 1).

An inverted T-shaped incision was made in the pericardium to suspend the heart in a pericardial cradle. In total, five pericardial stay sutures were applied. Two of these stay sutures were placed cranially and medially on the left pericardium to access the LA for catheterization (Figure 2).

The tip of the right atrial appendage was secured by a tie (Ethibond Excel No 2, Ethicon, Norderstedt, Germany) for ease of manipulation. The SVC’s extrapericardial part was dissected circumferentially, and the azygos vein was divided between transfixing sutures (6-0 Prolene, Ethicon, Norderstedt, Germany). Also, the SVC’s intrapericardial part was circumferentially dissected, ensuring that no veins draining into the right lateral aspect of the superior cavoatrial junction were torn (Figure 3).

The Ao and the MPA were separated. A 4 mm Mersilene tie (Ethicon, Norderstedt, Germany) was placed around the proximal MPA, enhancing access to the Ao and later banding of the MPA in the BDCPC group. The RPA was also dissected circumferentially including the superior lobe branch and the truncus intermedius (Figure 4).

A tie (Ethibond No 2, Ethicon, Norderstedt, Germany) was placed around the IVC and a vessel loop (Mediloop Mini, Neuromedex, Hamburg, Germany) around the hemiazygos vein draining into the coronary sinus [18] (Figure 5). Only in the BDCPC group was the hemiazygos vein occluded during cardioplegic arrest ro prevent cardioplegia wash-out.

A 3.5 Fr. pressure monitoring catheter (Vygon, Ecouen, France) was inserted into the SVC via a purse-string suture (6-0 Prolene, Ethicon, Norderstedt, Germany). Right-ventricular and left-atrial pressure were measured using a 3 Fr. pressure monitoring line (Medtronic, Minneapolis, MN, USA), respectively. Flow probes (Transonic COnfidence flowprobe, ADInstruments, Colorado Springs, CO, USA) were placed around the Ao and the SVC to assess CO and SVC flow. Normothermic CPB was established in both groups via an aortic cannula (14 Fr. Medtronic DLP, Minneapolis, United States), an angled venous cannula for SVC (16 Fr. Edwards Lifescience, Unterschleissheim, Germany) and IVC (20 Fr. Edwards Lifescience, Unterschleissheim, Germany), respectively. In both groups, the anastomosis was constructed identically. On the beating, heart the SVC was occluded close to the SVC cannula by a further vessel loop (Mediloop Mini, Neuromedex, Hamburg, Germany). The SVC’s 12 o’clock position was marked by a stitch (6-0 Prolene, Ethicon, Norderstedt, Germany) to ensure correct alignment of the anastomosis. The superior cavoatrial junction was clamped, and the SVC was detached from the heart. The atrial stump was oversewn by a running suture line (6-0 Prolene, Ethicon, Norderstedt, Germany). The RPA was occluded at its origin by a vessel loop. This loop helped to expose the RPA and remained occluded in the UDCPC group. Whereas in the BDCPC group this vessel loop was removed after completing the second anastomosis. The RPA’s branches were occluded using a vessel loop and a Yasargil clip (Geister, Tuttlingen, Germany). A 12 mm horizontal incision was carried out at the RPA’s cranial aspect (Figure 6), and a bevelled 12 mm ePTFE conduit (Gore-Tex stretch vascular graft, W.L. Gore and Associates Inc., Flagstaff, AZ, USA) (Figure 7) was anastomosed using a running suture line (6-0 Prolene, Ethicon, Norderstedt, Germany) (Figure 8).

Only in the BDCPC group, thereafter, was cardioplegic arrest achieved by aortic cross-clamping and administration of 30 mL/kg bodyweight cold (4 °C) HTK Bretschneider solution (Custodiol, Dr. Franz Köhler Chemie GmbH, Bensheim, Germany) via an aortic root cannula (5 Fr., Medtronic, Minneapolis, MN, USA). A low horizontal atriotomy was performed at the level of the caval axis. The fossa ovalis was exposed using two suction tips (Aesculap, Tuttlingen, Germany) (Figure 9). The membrane within the fossa ovalis was completely resected and an incision opening the layer between the coronary sinus and the LA was carried out in order to enlarge the interatrial communication (Figure 10 and Figure 11). Finally, an interatrial communication of 8 × 12 mm was achieved. The atriotomy was closed by a running suture line (6-0 Prolene, Ethicon, Norderstedt, Germany), and the heart was de-aired via the cardioplegia cannula.

In both groups, the Gore-Tex conduit was shortened to a length of 29 to 31 mm, and an end-to-end anastomosis between the conduit and the transected SVC was performed (6-0 Prolene, Ethicon, Norderstedt, Germany) (Figure 12). After 2 min of pulmonary recruitment (20 mbar peak inspiratory pressure (PIP), 10 mbar positive end expiratory pressure (PEEP), 15 breaths per minute), full ventilation was resumed and, the heart was weaned off CPB using adrenaline 0.05 µg/kg × min.

### 2.4. Cardiopulmonary Bypass Set-Up

A Stöckert S3 heart–lung machine and heater cooler were used (LivaNova, Munich, Germany). A membrane oxygenator with integrated 32 µm arterial line filter (Capiox FX 15, Terumo, Tokyo, Japan) and a sterile tubing set were used. Normothermic CPB was established with a pump flow of 75% baseline CO during partial bypass and 130% baseline CO during total bypass. The pump speed was adjusted to maintain a MAP between 55 and 75 mmHg and to avoid the use of vasopressor medication during CPB. During extracorporeal circulation, mechanical ventilation was set to 5 breaths per minute at 10 mbar PIP and a 5 mbar PEEP. The pump priming consisted of 500 mL citrated porcine full blood (Xebios Diagnostic GmbH, Duesseldorf, Germany), 500 mL isotonic, iso-oncotic crystalloid solution (Fresenius Kabi, Bad Homburg, Germany) and 2000 IU heparin (B. Braun Melsungen AG, Melsungen, Germany). For anticoagulation, 6000 IU heparin were administered via the central venous line. After separation from CPB, 8000 IU of protamine reversed the heparin effect. Finally, all cannulas were removed.

### 2.5. Data Acquisition

Two measurements were taken in both groups. After all lines and flow probes were implanted, the animals recovered for 30 min and the baseline measurements were taken. In both groups, 60 min after creation of the cavopulmonary connection, the second set of measurements was taken. In the BDCPC group, the animals were weaned off CPB and protamine was administered. Thereafter, the MPA banding was gradually tightened over a 20 min period by applying ligating clips (Horizon, Teleflex Medical, Morrisville, NC, USA) on the Mersilene band while hemodynamics, blood gases, and echocardiography were monitored to create a non-pulsatile flow trace in the SVC. Additionally, echocardiography confirmed the presence of antegrade pulmonary blood flow through the pulmonary valve and a right-to-left shunt via the created atrial communication. Intracardiac blood flow, TAPSE, and LVEF were assessed echocardiographically using a S8-3 transducer on a model iE33 device (Philips Medical Systems, Hamburg, Germany).

The blood flow measurements in the Ao and the cavopulmonary connection were carried out using a TS402 flow sensor and a TS420 perivascular flowmeter module (Transonic System Inc., New York, NY, USA). All pressure measurements were made using the MLT844 physiological pressure transducer (ADInstruments Ltd., Sydney, Australia). All pressure and flow measurements were recorded using Powerlab C (ADInstruments Ltd., Sydney, NSW, Australia).

### 2.6. Data Analysis

Continuous data are expressed as median (interquartile range) or mean ± standard error of mean, depending on data distribution. Normal distribution was tested by the Shapiro–Wilk test. Three measurements, separated by two minutes, were taken at baseline and one hour after separation from CPB and averaged for statistical evaluation, respectively. The statistical significance of changes from baseline values within each parameter was tested with a paired *t*-test. Differences between both groups were analysed by an unpaired *t*-test. Statistical significance was accepted at *p* ≤ 0.05. Statistical analysis was performed with GraphPad Prism 10 (GraphPad Software, Boston, MA, USA).

## 3. Results

All animals survived the surgery and the following one-hour observation period. All constructed anastomoses remained patent, without distortion and kinking, as confirmed by direct inspection after the experiment. All animals remained on inotropic support throughout the observation period. Each group included two female animals. There was no significant difference in any investigated parameter.

In the BDCPC group, initially, after coming off CPB, all animals were stable. However, the pressure in the cavopulmonary connection was 22.8 ± 3.2 mmHg with a clearly visible pulsatile pressure tracing. The gradual intensification of the MPA banding over a 20 min period led to a BDCPC with a non-pulsatile cavopulmonary pressure trace. This resulted in an impaired cardiopulmonary state. Subsequently, CO dropped from 3.15 ± 0.21 to 2.17 ± 0.19 L/min (*p* < 0.01) leading to a significant reduction in the MAP from 80.8 ± 3.7 to 49.3 ± 7.3 mmHg (*p* = 0.01) (Table 1 and Table 2). Especially, the RV function was impaired as indicated by a significant reduced TAPSE compared to baseline (*p* < 0.01) as well as to the UDCPC group (*p* < 0.01) (Table 2). Also, the RV pressure increased compared to baseline (Table 1). In contrast, the LV function was not compromised, as demonstrated by an increased LVEF and an unaltered LAP (Table 1 and Table 2). Echocardiography in this group revealed an unrestrictive right-to-left shunt on atrial level in all animals, and a patent right ventricle to pulmonary artery connection with an accelerated flow (3.3 ± 0.3 m/s). Even after application of a tight pulmonary artery band, the SVC pressure remained significantly increased compared to baseline (*p* = 0.03) and the UDCPC group (*p* = 0.04) (Table 1). As a consequence of the circulatory impairment, a significant lactic acidosis resulted only in this group (*p* = 0.01) (Table 3). Furthermore, the blood gas analysis revealed a significantly reduced gas exchange with a significantly reduced arterial partial pressure of oxygen (pO_2_) (*p* < 0.01) and increased pCO_2_ compared to baseline (*p* = 0.03) and the unidirectional cavopulmonary connection group (*p* = 0.03) (Table 3). Accordingly, the arterial saturation was significantly reduced from 95.8 ± 1.1 to 60.9 ± 10.4% (Table 3), although the fraction of inspired oxygen was significantly increased to 100 (IQR 40)% compared to baseline of 21 (IQR 1)%.

In contrast, in the unidirectional cavopulmonary connection group the cardiopulmonary parameters indicated a stable condition. Although the blood flow trace in the cavopulmonary connection was also non-pulsatile, its pressure level was significantly elevated compared to baseline (*p* < 0.01) (Table 1).

Only a mild tachycardia was present (Table 1), and on echocardiography, the right ventricle showed mild signs of right ventricular strain (Table 1 and Table 2). In this group, the left ventricular function remained also unaffected (Table 1 and Table 2). The blood gas analysis confirmed the stable cardiopulmonary condition in the unidirectional cavopulmonary connection group (Table 3).

The conduct of CPB resulted in unaltered electrolyte concentrations in both groups. A slightly elevated but significant hematocrit level one hour after termination of CPB resulted (*p* = 0.02). However, one unit containing 500 mL fresh porcine whole blood was required for the pump priming in all experiments. In the UDCPC group, pump speed was significantly higher compared to the BDCPC group, resulting in an increased mean arterial blood pressure (Table 4). However, in both groups, the mean arterial blood pressure during extracorporeal circulation remained within the range set by the perfusion protocol (Table 4).

## 4. Discussion

For the purpose of investigating the effect of unilateral diaphragmatic palsy on pulmonary blood flow in a univentricular animal model, we intended to establish a reproducible large animal model. In the upcoming experimental series, we are then going to create unilateral palsy by transecting the right phrenic nerve. The main focus of the currently established model was the creation of a non-pulsatile pulmonary blood flow in the RPA without any phrenic nerve affection. Another requirement for studying the effects of unilateral phrenic nerve palsy on ipsilateral pulmonary artery blood flow is achieving a stable cardiopulmonary condition after establishing the cavopulmonary connection. Hemodynamic stability allows further extensive cardiopulmonary research in an acute setting. Previously, a cavopulmonary connection has been successfully realised experimentally in dogs [19,20,21,22,23], pigs [13,14,15,17], lambs [24,25], and rabbits [26]. Even a transcatheter approach for a cavopulmonary connection has been published [16]. Nowadays, it is nearly impossible to conduct animal research on dogs and lambs due to ethical constrains raised by the relevant authorities in Germany. A rabbit’s body size is too small to accommodate the unilateral diaphragmatic pacing device. Therefore, under strict control by the relevant state authorities, we are still able to conduct large animal research in porcine models. So, we initially attempted to employ the porcine model published by Henaine and coworkers [14,27]. They constructed a superior cavopulmonary connection without the use of CPB or even the insertion of a shunt from the SVC to the right atrium. In our preliminary attempts, the animals did not tolerate clamping of the SVC for the time required to finalise the dissection and construct the anastomosis. So, we inserted an angled venous cannula in the SVC and the right atrial appendage to relieve the superior inflow congestion caused by prolonged SVC clamping. The cannulas were connected by a straight 3/8 inch connector [28,29,30]. Heparin (100 I.U.) was administered intravenously. After tightening a tourniquet around the SVC and detaching the SVC right at its junction from the heart, the RPA, and the SVC, including its tributaries, were fully mobilised. The RPA branches were occluded using Yasargil clips, and the RPA was clamped. A horizontal incision at the RPA’s cranial aspect was performed. The SVC did not reach down to the incision to fully complete the anastomosis in all five attempts. The inability to reach the RPA without extension is consistent with all other groups that have previously performed a cavopulmonary connection in pigs, except for Henaine et al. [13,15,17,31,32,33,34,35]. Alternatively, these research groups aimed for direct anastomosis of the SVC to the MPA [15], tried reaching the MPA via a tube graft extension [13,17,31,32,34,35], or tried creating an end-to-end-anastomosis between the superior caval vein and the RPA with a Gore-Tex tube graft extension [33]. Since we aim for the SVC’s more naturally route straight downward to the right pulmonary artery, we decided to extend the anastomosis by an autologous pericardial patch. This leads to a straight connection resulting in undisturbed laminar blood flow. Directing the SVC to the MPA may more likely result in kinking due to the relevant SVC’s leftward shift. We then attempted an anastomosis between the SVC’s posterior wall and the cranial aspect of the horizontal arteriotomy in the RPA. A patch augmentation of the remaining defect across the anterior aspect of the SVC’s margin and the anterior aspect of the RPA incision was performed. The required pull on the heart for exposure was not tolerated in three animals, so these animals succumbed before the anastomosis was completed. The anastomosis was completed in two animals. In one of these animals, the suture line partly tore through the RPA tissue, leading to severe haemorrhage. The other animal with a completed anastomosis succumbed minutes after releasing all clamps due to circulatory failure, although high inotropic medication was administered. Therefore, in our hands, the off-pump procedure represents no viable option compared to a procedure supported by extracorporeal circulation, although the use of the extracorporeal circulation leads to various other well-known side effects and requires the addition of blood for the pump priming. However, in this particular context of a very caudally displaced pulmonary artery below the level of the right superior pulmonary vein, its use seems essential to achieve cardiopulmonary stability. Additionally, we opted for the use of a tube graft extension for the cavopulmonary connection between the right pulmonary artery and the superior vena cava. This approach appears feasible also to other research groups [33,34,35] and preserves the favourable geometry between SVC and RPA, leading to laminar blood flow in the constructed anastomosis. From our point of view, for the upcoming experiments on the effect of diaphragmatic palsy on pulmonary artery blood flow in a univentricular heart model, achieving an uncompromised connection between the RPA and SVC is of utmost importance.

In our set-up, the unidirectional cavopulmonary connection was created without atrial communication. This configuration proved to be superior in regard to cardiopulmonary stability compared to the bidirectional cavopulmonary connection with limited antegrade pulmonary blood flow. For our intended purpose of studying the effects of a unilateral diaphragmatic pacemaker on pulmonary blood flow in a univentricular heart setting, a non-pulsatile pulmonary blood flow on the studied right side is essential. Therefore, we were compelled to tighten the MPA banding in the BDCPC group to an extent that caused substantial right ventricular afterload increase. This caused acute right heart failure leading to a low cardiac output state. The competent tricuspid valve in all animals prevented right ventricular decompression. In consequence, right ventricular failure led to increased right atrial pressure. The created atrial communication allowed partial right atrial decompression leading to some degree of improved left sided filling. However, the shunt volume did not suffice to achieve a near-normal cardiac output. There are potential alternatives that have not been studied yet. It seems possible to create a relevant tricuspid regurgitation or a ventricular septal defect to decompress the right ventricle. The extent of both these measures is extremely difficult to calibrate. The chosen ePTFE tube graft extension lacks growth potential. Therefore, the use of a dilatable tube graft offers the potential for size adaptation. Currently, a dilatable ePTFE tube graft (Exgraft, PECA Labs, Pittsburgh, PA, USA) is available on the market, allowing for a substantial increase in diameter of up to 40% of the original diameter [36]. Using this alternative conduit, our model might also be suitable for mid- to long-term outcome studies.

The present study is facing several limitations. Primarily, the subjects were healthy pigs with a biventricular heart and no previous surgical intervention. In contrast, a superior cavopulmonary anastomosis is usually performed in children with a functionally single ventricle. In most cases, the creation of a superior cavopulmonary connection is executed in a redo procedure. Furthermore, the present model investigates only the acute changes after creation of a superior cavopulmonary connection. Our current approach enables the formulation of a preliminary statement regarding the newly established haemodynamics in the acute setting. Further research is required to study the effect over an extended time period. This is particularly pertinent given the significant influence of spontaneous breathing on the circulatory physiology of the superior cavopulmonary anastomosis. Consequently, further experimentation should aspire to establish a chronic animal model based on the findings of this study.

This long-term follow up may allow for the assessment of anastomosis quality and the resulting changes in pulmonary artery flow patterns, as short-term assessments are not very relevant to its long-term behaviour.

## 5. Conclusions

The use of CPB and a conduit extension allowed the creation of a reproducible cavopulmonary connection. Cardiopulmonary stability was achieved by anastomosing the SVC to the excluded RPA, resulting in a unidirectional cavopulmonary connection. The creation of a bidirectional cavopulmonary connection with limited antegrade pulmonary blood flow and an unrestrictive atrial septal communication resulted in a less stable hemodynamic condition. This model offers a promising basis for further studies regarding cavopulmonary connections and their associated physiology. The reproducibility of this model may aid in future research on partial cavopulmonary connections.

## Figures and Tables

**Figure 1 biomedicines-13-00918-f001:**
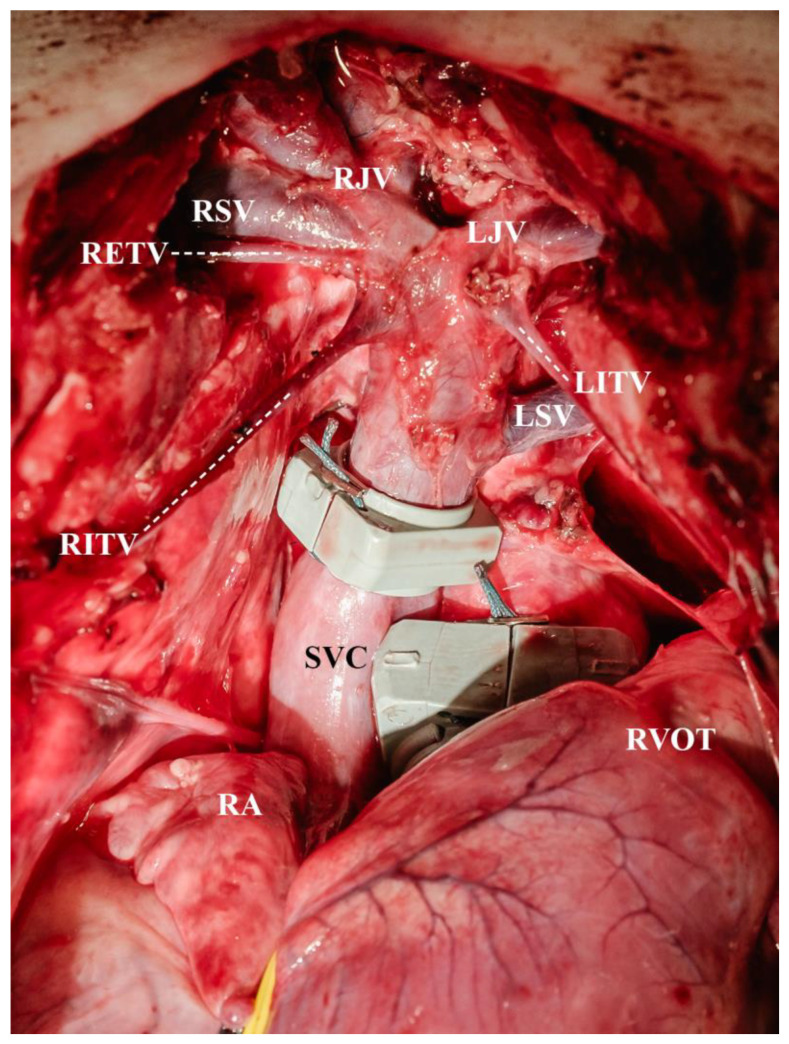
Circumferentially dissected superior vena cava including the dissected tributaries (LITV = left internal thoracic vein, LJV = left jugular vein, LSV = left subclavian vein, RA = right atrium, RETV and adjacent dotted line = right external thoracic vein, RITV and adjacent dotted line = right internal thoracic vein, RJV = right jugular vein, RSV = right subclavian vein, RVOT = right ventricular ouflow tract, SVC = superior vena cava).

**Figure 2 biomedicines-13-00918-f002:**
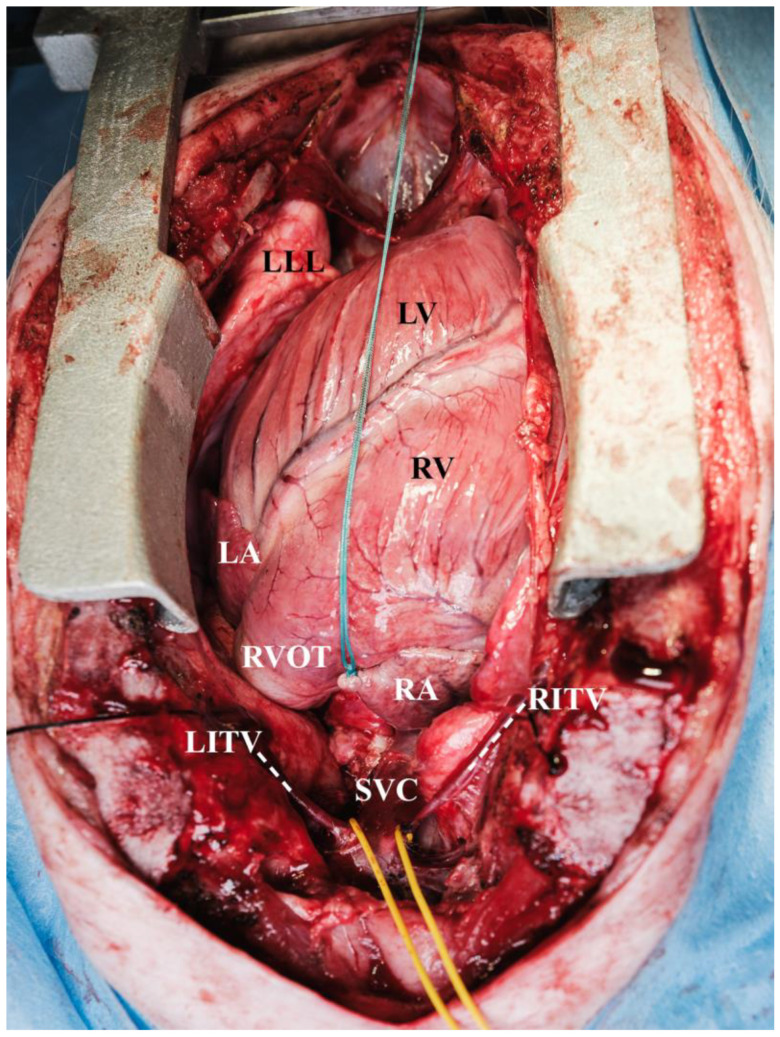
Heart suspended in the pericardial cradle (LA = left atrium (auricle), LITV and adjacent dotted line = left internal thoracic vein, LLL = left lower lobe of the lung, LV = left ventricle, RA = right atrium, RITV and adjacent dotted line = right internal thoracic vein, RV = right ventricle, RVOT = right ventricular outflow tract, SVC = superior vena cava).

**Figure 3 biomedicines-13-00918-f003:**
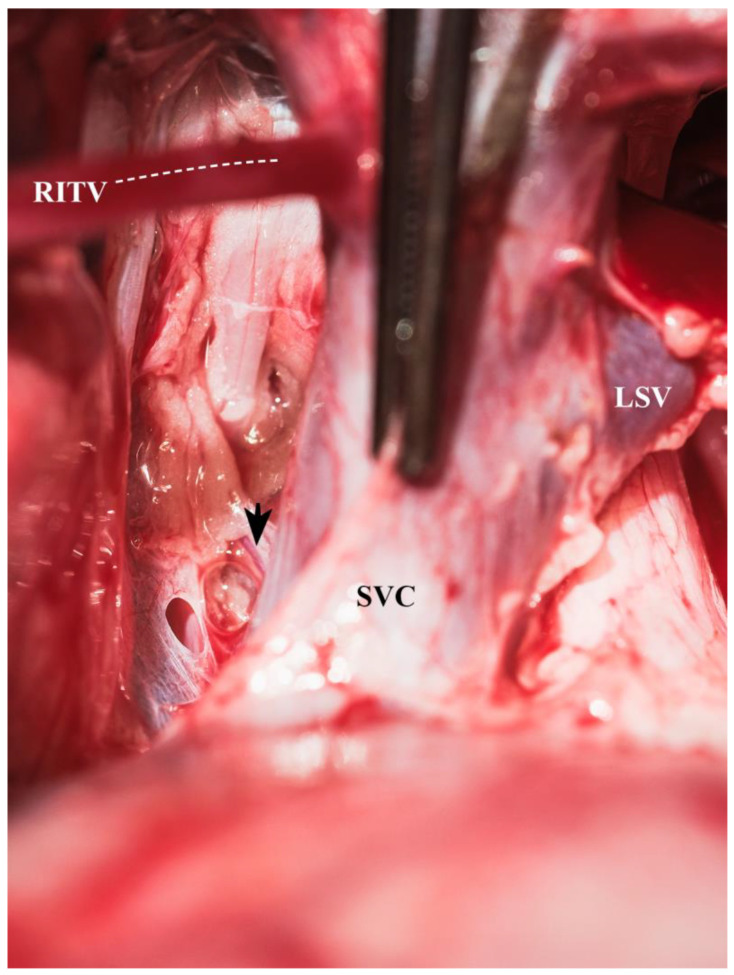
Mediastinal vein draining into the superior vena cava (arrowhead pointing to a particular vein draining into the SVC close to the cavoatrial junction; LSV = left subclavian vein, RITV and adjacent dotted line = right interior thoracic vein, SVC = superior vena cava).

**Figure 4 biomedicines-13-00918-f004:**
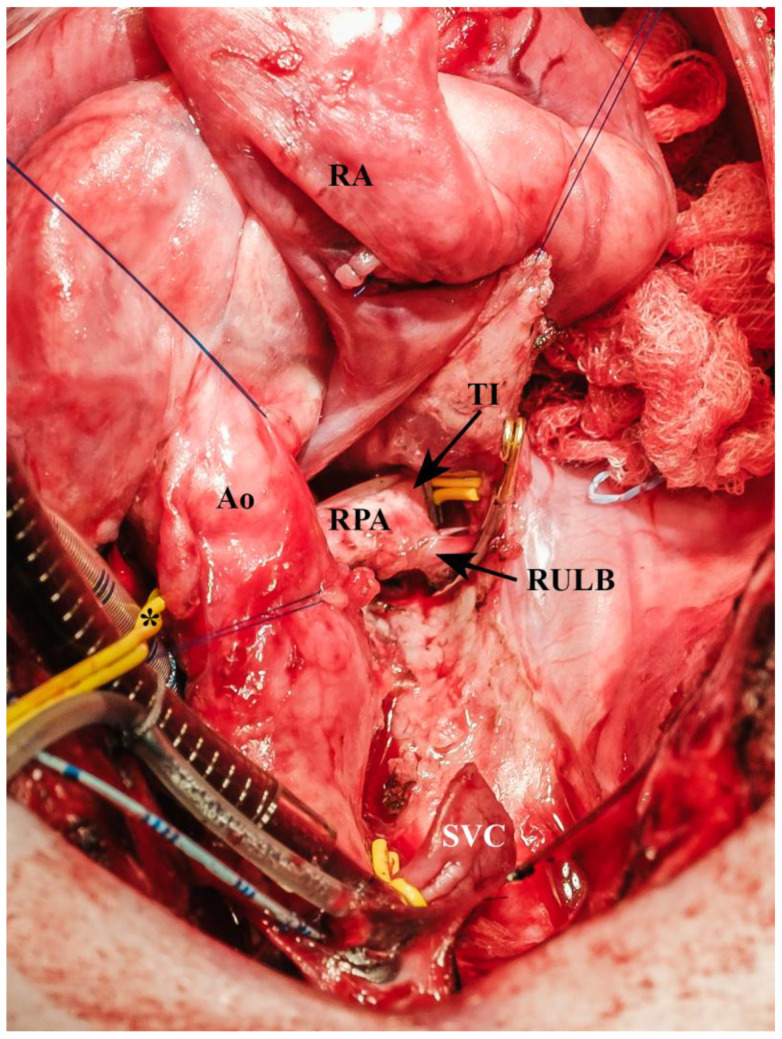
Dissected right pulmonary artery including the superior lobe branch and the Truncus intermedius branch (* = vessel loop around proximal right pulmonary artery, Ao = aorta, RA = right atrium, RPA = right pulmonary artery, RULB = right upper lobe branch of pulmonary artery (occluded by a Yasargil clip), SVC = superior vena cava stump (occluded by a vessel loop and clip), TI = truncus intermedius (occluded by a vessel loop and clip)).

**Figure 5 biomedicines-13-00918-f005:**
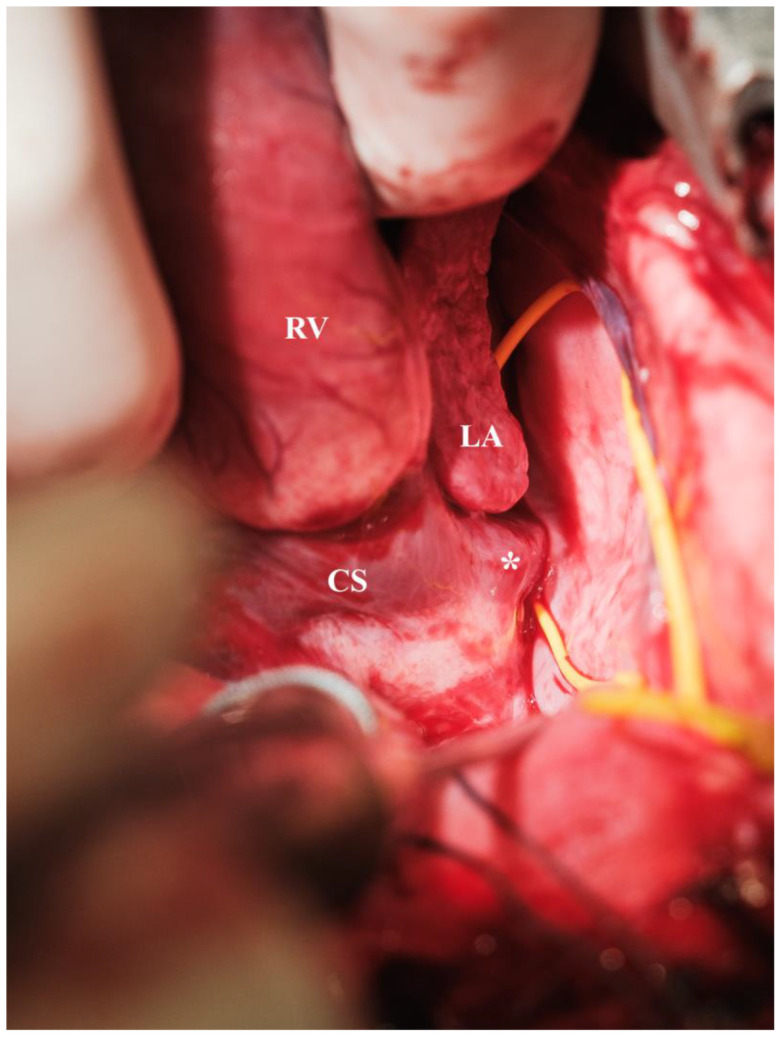
Hemiazygos vein encircled by a vessel loop (* = hemiazygos vein, CS = coronary sinus, LA = left atrium, RV = right ventricle).

**Figure 6 biomedicines-13-00918-f006:**
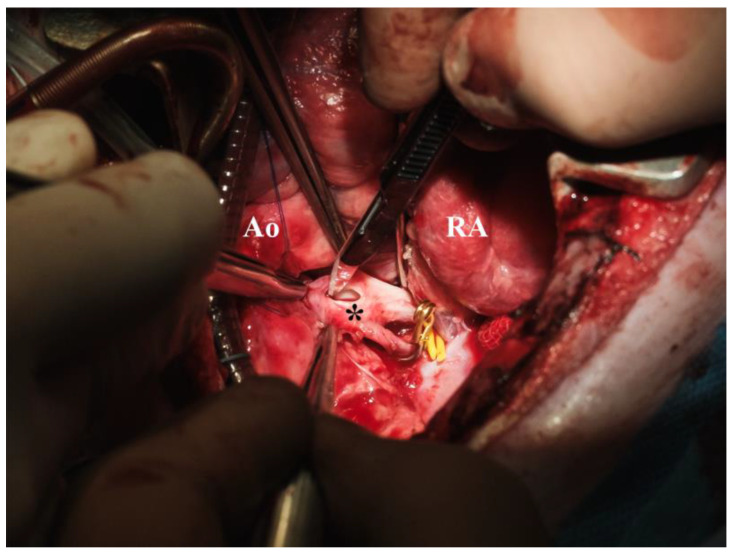
Right pulmonary artery incision (* = right pulmonary artery, Ao = aorta, RA = right atrium).

**Figure 7 biomedicines-13-00918-f007:**
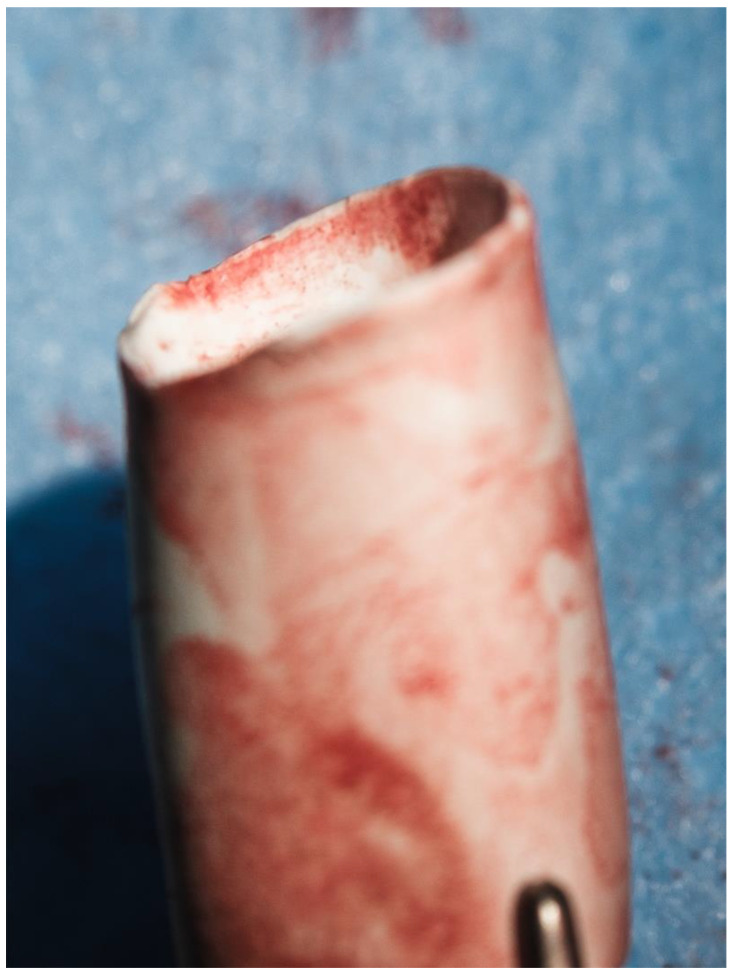
Bevelled 12 mm ePTFE conduit.

**Figure 8 biomedicines-13-00918-f008:**
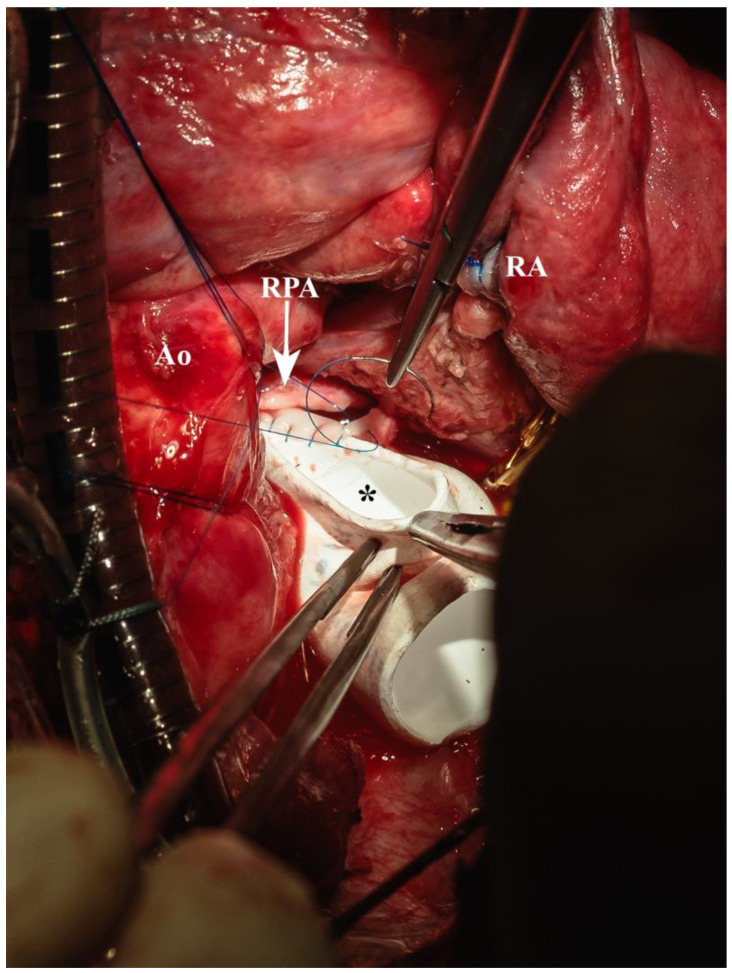
Proximal anastomosis in running suture line (* = ePTFE conduit, Ao = aorta, RA = right atrium, RPA = right pulmonary artery).

**Figure 9 biomedicines-13-00918-f009:**
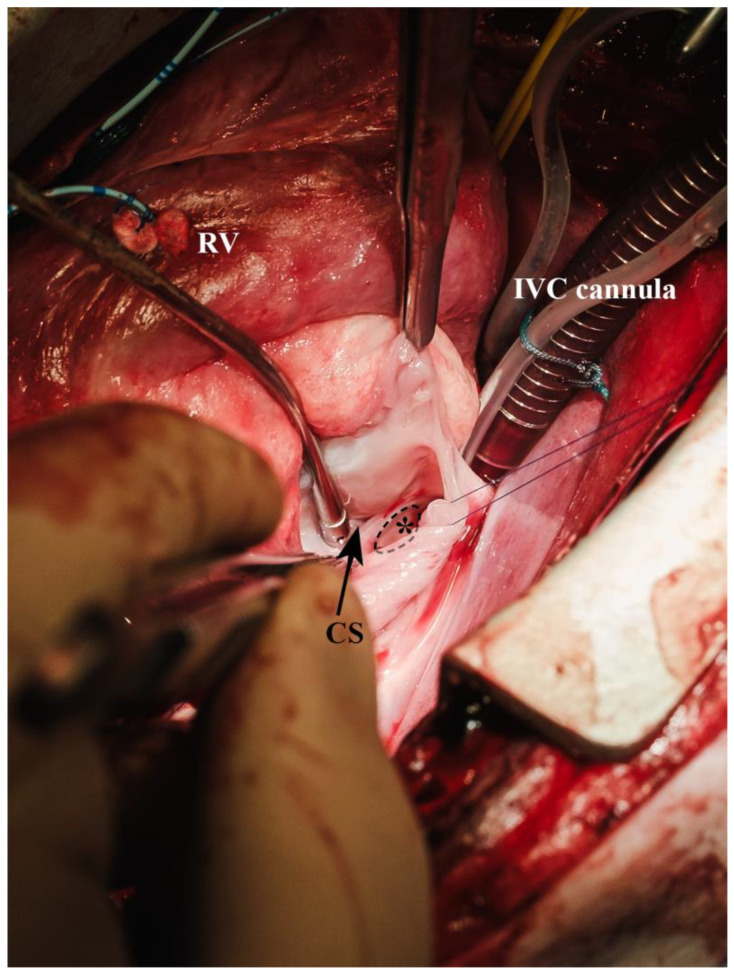
Exposition of the fossa ovalis via vertical atriotomy (* = fossa ovalis (dashed outline), CS = coronary sinus, IVC cannula = inferior vena cava cannula, RV = right ventricle with pressure catheter).

**Figure 10 biomedicines-13-00918-f010:**
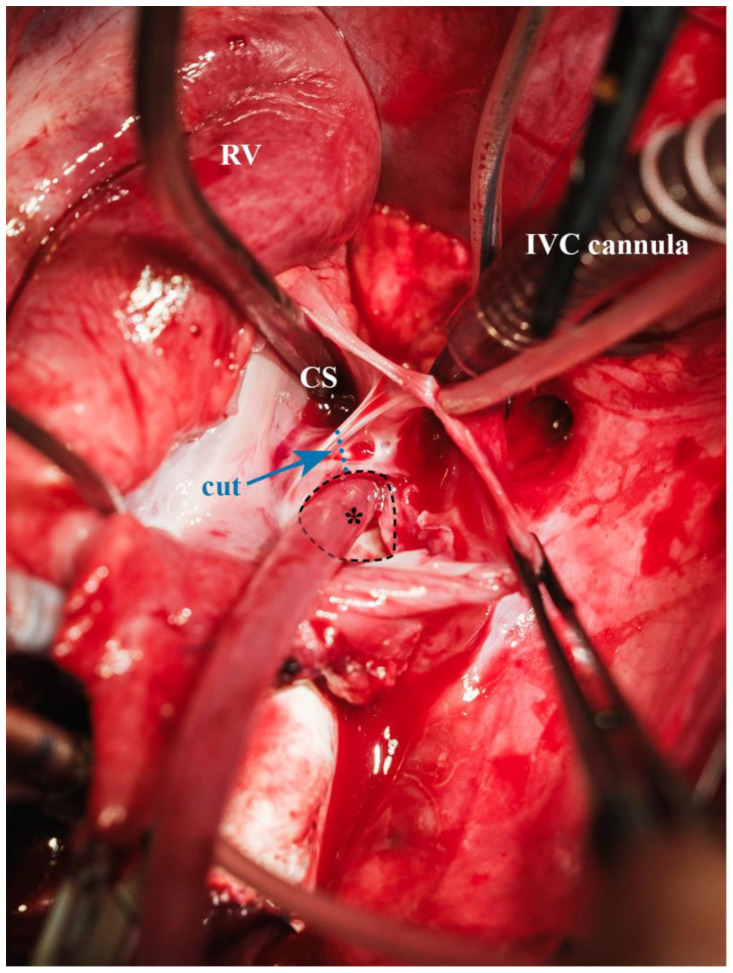
Completely resected membrane within the fossa ovalis (* = suction tip in atrial communication (dashed outline), CS = suction tip in coronary sinus, blue arrow = site of atrial communication enlargement (blue dotted line), IVC cannula = inferior vena cava cannula, RV = right ventricle).

**Figure 11 biomedicines-13-00918-f011:**
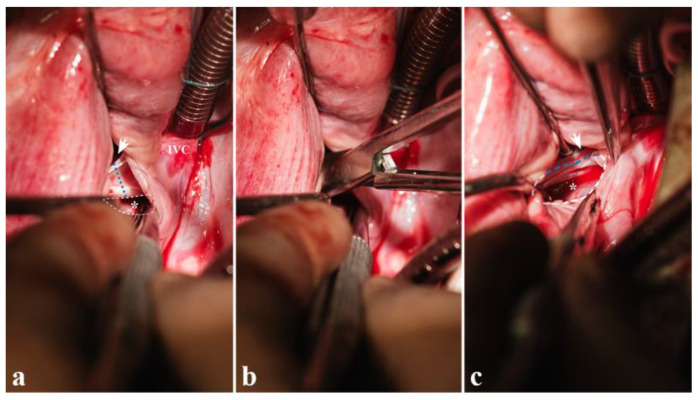
Resection of the coronary sinus floor: (**a**) intact coronary sinus floor, (**b**) resection with scissors, (**c**) completed atrial communication (* = atrial communication (dashed outline), arrow head in coronary sinus, blue dotted line = site of enlarging cut, IVC = inferior vena cava).

**Figure 12 biomedicines-13-00918-f012:**
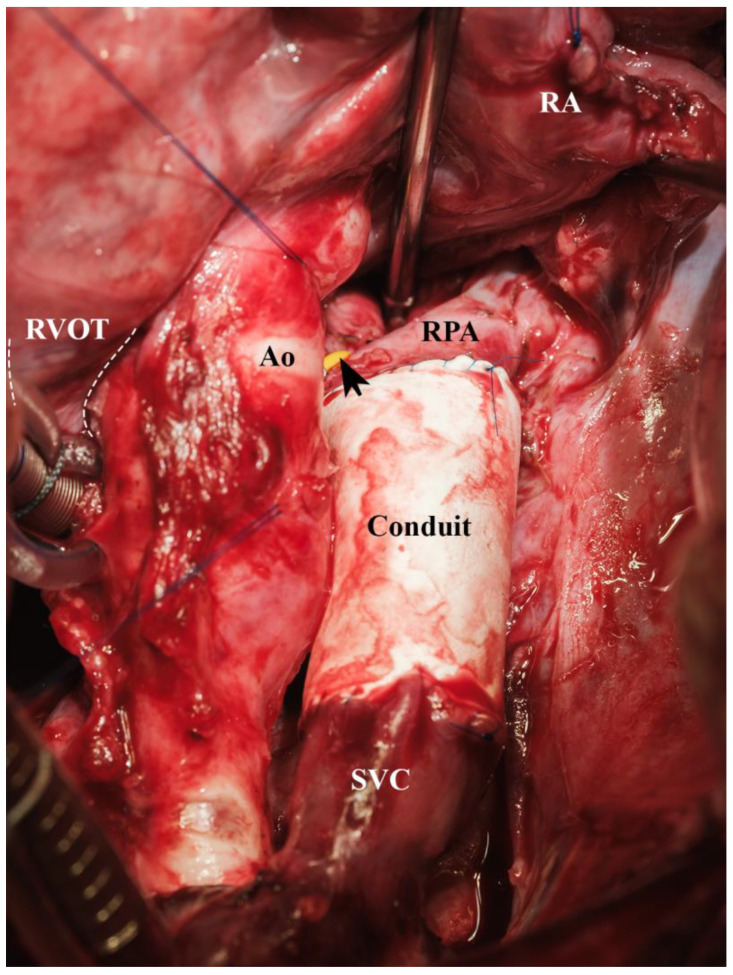
Superior cavopulmonary connection established by a 12 mm ePTFE conduit extension (arrow on right pulmonary artery trunk, occluded via vessel loop and clip, Ao = aorta, Conduit = bevelled 12 mm ePTFE conduit, RA = right atrium, RPA = right pulmonary artery, RVOT = right ventricular outflow tract (trajectory indicated by dotted lines), SVC = superior vena cava).

**Table 1 biomedicines-13-00918-t001:** Hemodynamic assessment.

	Bidirectional CPC		Unidirectional CPC	
	Baseline	60 min Post CPC	Baseline	60 min Post CPC
HR [min^−1^]	94 ± 5	104 ± 7	91 ± 3	113 ± 6 *
MAP [mmHg]	80.8 ± 3.7	49.3 ± 7.3 *	71.9 ± 2.3	62.4 ± 2.8
SVCP [mmHg]	7.5 ± 1.3	17.0 ± 1.0 *	7.3 ± 0.8	13.9 ± 0.8 *†
IVCP [mmHg]	8.2 ± 0.8	12.0 ± 1.5	7.7 ± 0.7	8.6 ± 0.9
LAP [mmHg]	7.6 ± 1.0	7.7 ± 2.1	9.2 ± 0.8	9.3 ± 0.8
RVSP [mmHg]	35 ± 3	46 ± 7	33 ± 2	41 ± 3 *

Data presented as mean ± standard error of mean. * = significant difference within the group (*p* < 0.05), † = significant difference between groups (*p* < 0.05). Bidirectional CPC with atrial septal communication and MPA banding (*n* = 6). Unidirectional CPC without atrial septal communication (*n* = 8). CPC = cavopulmonary connection, HR = heart rate, MAP = mean arterial blood pressure, SVCP = superior vena cava blood pressure, IVCP = inferior vena cava blood pressure, LAP = left atrial pressure, RVSP = right ventricular systolic pressure.

**Table 2 biomedicines-13-00918-t002:** Cardiac function.

	Bidirectional CPC		Unidirectional CPC	
	Baseline	60 min Post CPC	Baseline	60 min Post CPC
CO [L/min]	3.15 ± 0.21	2.17 ± 0.19 *	2.97 ± 0.22	2.6 ± 0.25
F(SVC) [L/min]	1.07 ± 0.19	0.43 ± 0.07 *	0.97 ± 0.14	0.5 ± 0.1 *
LVEF [%]	68 ± 4	76 ± 5	74 ± 2	71 ± 3
TAPSE	12 ± 0.7	5 ± 0.7 *	13 ± 0.8	9 ± 0.4 *†

Data presented as mean ± standard error of mean. * = significant difference within the group (*p* < 0.05), † = significant difference between groups (*p* < 0.05). Bidirectional CPC with atrial septal communication and MPA banding (*n* = 6). Unidirectional CPC without atrial septal communication (*n* = 8). CO = cardiac output, F(SVC) = superior vena cava flow, LVEF = left ventricular ejection fraction, TAPSE = tricuspid annular plane systolic excursion.

**Table 3 biomedicines-13-00918-t003:** Arterial blood gas analysis.

	Bidirectional CPC		Unidirectional CPC	
	Baseline	60 min Post CPC	Baseline	60 min Post CPC
pH	7.45 ± 0	7.3 ± 0 *	7.45 ± 0	7.41 ± 0 †
pO_2_ [mmHg]	91.0 ± 6.3	46.9 ± 6.9 *	91.8 ± 4.2	151.8 ± 10.0 *†
SO_2_ [%]	95.8 ± 1.1	60.9 ± 10.4 *	96.0 ± 1.0	99.4 ± 0.2 *†
pCO_2_ [mmHg]	41.3 ± 1.6	52.4 ± 3.1 *	40.8 ± 1.2	44.4 ± 1.7 †
Hct [%]	24.0 ± 0.6	28.3 ± 1.7 *	24.6 ± 1.8	26.0 ± 2.3
Lactate [mmol/L]	0.82 ± 0.09	4.36 ± 0.96 *	1.37 ± 0.18 †	2.17 ± 0.45 †

Data presented as mean ± standard error of mean. * = significant difference within the group (*p* < 0.05), † = significant difference between groups (*p* < 0.05). Bidirectional CPC with atrial septal communication and MPA banding (*n* = 6). Unidirectional CPC without atrial septal communication (*n* = 8). pH = potential hydrogenii, pO_2_ = partial pressure of oxygen, SO_2_ = oxygen saturation, pCO_2_ = partial pressure of carbon dioxide, Hct = hematocrit.

**Table 4 biomedicines-13-00918-t004:** Cardiopulmonary bypass characteristics.

	Bidirectional CPC	Unidirectional CPC
CPB flow [L/min]	2.57 ± 0.47	3.30 ± 0.22 *
MAP [mmHg]	62 ± 11	75 ± 10
CPB time [min]	156 ± 36	133 ± 16
pH	7.44 ± 0.05	7.41 ± 0.04
pO_2_ [mmHg]	435 ± 77	407 ± 43
pCO_2_ [mmHg]	41 ± 3	49 ± 5 *
Hct [%]	22 ± 1	21 ± 3

Data presented as mean ± standard error of mean. Bidirectional CPC with atrial septal communication and MPA banding (*n* = 6). Unidirectional CPC without atrial septal communication (*n* = 8). CPB flow = cardiopulmonary bypass main pump flow, MAP = mean arterial blood pressure, CPB time= total time on cardiopulmonary bypass, pH = potential hydrogenii, pO_2_ = partial pressure of oxygen, pCO_2_ = partial pressure of carbon dioxide, Hct = hematocrit. * = significant difference within the group (*p* < 0.05).

## Data Availability

Data are contained within the article.

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
