# Peer review of "Establishment and Hemodynamic Assessment of the Superior Cavopulmonary Anastomosis in a Reproducible Porcine Model"

_biomedicines, 2025, doi:10.3390/biomedicines13040918_

Round 1
Reviewer 1 Report
Comments and Suggestions for Authors
- In the Introduction area the authors state that “In order to effectively study the potential treatment option of a synchronized unilateral diaphragmatic pacemaker r, we first intended to establish a model that mimics the cardiopulmonary status of a univentricular heart patient with non-pulsatile pulmonary artery blood flow.”. The authors are kindly asked to put some information on the full concept and possible novelty of their work in the Background section of the Abstract so that the audience is fully aware of the full perspective.
- The authors are also kindly asked to give the date (and, if possible, the number) of the ethics committee approval.
- The authors are also asked to mention/outline whether this animal experiment complies with the “3Rs alternatives” rule, referring to the replacement, reduction, and refinement of animals used in research, teaching, testing, and exhibition.
- Congratulation to the authors who have performed technically challenging procedure in porcine models. It would be important to understand the authors’ overall experience in cardiac and particularly experimental cardiac surgery on pigs in order for the audience to understand the anticipated reproducibility of the complex experiment.
- The authors provided p-values neither in abstract on in the manuscript. The authors are kindly asked to provide this information.
- In the “Discussion section” the authors mention once again that “For the purpose of investigating the effect of unilateral diaphragmatic palsy on pulmonary blood flow in a univentricular animal model, we intended to establish a reproducible large animal model.” The authors are kindly asked to provide information whether there were any such neurological complications in their acute experiment.
- The authors are kindly asked to describe the limitations of the study more clearly in the Discussion area.
Author Response
Reviewer 1:
1. In the Introduction area the authors state that “In order to effectively study the potential treatment option of a synchronized unilateral diaphragmatic pacemaker, we first intended to establish a model that mimics the cardiopulmonary status of a univentricular heart patient with non-pulsatile pulmonary artery blood flow.”. The authors are kindly asked to put some information on the full concept and possible novelty of their work in the Background section of the Abstract so that the audience is fully aware of the full perspective.
We added the required information on page 1, line 16 to 24.
2. The authors are also kindly asked to give the date (and, if possible, the number) of the ethics committee approval.
The ethics committee approval’s number has been provided in the initially submitted version on page 19, line 426 to 428. However, we added the date of approval in the current version on page 22, line 516.
3. The authors are also asked to mention/outline whether this animal experiment complies with the “3Rs alternatives” rule, referring to the replacement, reduction, and refinement of animals used in research, teaching, testing, and exhibition.
We thank the reviewer for this valuable contribution and added a statement on page 3, line 90 to 92.
4. Congratulation to the authors who have performed technically challenging procedure in porcine models. It would be important to understand the authors’ overall experience in cardiac and particularly experimental cardiac surgery on pigs in order for the audience to understand the anticipated reproducibility of the complex experiment.
We amended the "Materials and Methods“ section accordingly on page 4, line 141 to 143.
5. The authors provided p-values neither in abstract on in the manuscript. The authors are kindly asked to provide this information.
We have changed the entire manuscript according to the reviewer‘s legitimate criticism and added p-values for all statistical analysis.
6. In the “Discussion section” the authors mention once again that “For the purpose of investigating the effect of unilateral diaphragmatic palsy on pulmonary blood flow in a univentricular animal model, we intended to establish a reproducible large animal model.” The authors are kindly asked to provide information whether there were any such neurological complications in their acute experiment.
We have clarified our intention of investigating the effect of phrenic nerve palsy on pulmonary blood flow in upcoming experiments utilising the currently established single ventricle model. This information is provided on page 19, line 400 to 403.
7. The authors are kindly asked to describe the limitations of the study more clearly in the Discussion area.
A limitation section was added to the “Discussion“ on page 21, line 479 to 493.
Reviewer 2 Report
Comments and Suggestions for Authors
I congratulate the authors for their research. I appreciate the practical aspect of the study as I have seen many complications of congenital heart diseases surgical treatment. Currently, there is a lack of standardised surgical techniques and this experiment is a step forward towards optimising palliative surgical care. I would appreciate a long term follow up of the model to assess the anastomosis, flow changes and pulmonary circulation as the short term assessment is not very relevant to its long term behaviour.
Author Response
Reviewer 2:
I congratulate the authors for their research. I appreciate the practical aspect of the study as I have seen many complications of congenital heart diseases surgical treatment. Currently, there is a lack of standardised surgical techniques and this experiment is a step forward towards optimising palliative surgical care. I would appreciate a long term follow up of the model to assess the anastomosis, flow changes and pulmonary circulation as the short term assessment is not very relevant to its long term behaviour.
We thank the reviewer for their constructive response and added a statement in the “Discussion“ section on page 21, line 491 to 493.
Reviewer 3 Report
Comments and Suggestions for Authors
The palliative surgery for the treatment of functionally univentricular heart malformations it is a very interesting idea.
I believe that the manuscript proposed for publication is particularly valuable, even if it is primarily addressed to cardiovascular surgeons and less to cardiologists.
However, I note the format of the work and also the extremely pertinent conclusion.
Thus, the research methodology is rigorously described, taking into account all current recommendations regarding a valuable original article.
Also, the use of intraoperative images significantly increases reader adherence and also the way of understanding the results.
The conclusion that the unidirectional cavopulmonary connection indicated a more stable cardiopulmonary condition it is particularly valuable, with possible implications for medical practice.
Regarding the References, I noticed the presence of only 4 newer citations from 2020, the rest are dominated by the oldest ones, from 1990 to 2010. Even though there are old extremely valuable articles and publications even today, I believe that the authors should expand the list of references with newer publications, this informations can be included in the Discussions chapter and also to do a comparative analysis with the results obtained in the current study.
Author Response
Reviewer 3:
The palliative surgery for the treatment of functionally univentricular heart malformations it is a very interesting idea.
I believe that the manuscript proposed for publication is particularly valuable, even if it is primarily addressed to cardiovascular surgeons and less to cardiologists. However, I note the format of the work and also the extremely pertinent conclusion.Thus, the research methodology is rigorously described, taking into account all current recommendations regarding a valuable original article.
Also, the use of intraoperative images significantly increases reader adherence and also the way of understanding the results. The conclusion that the unidirectional cavopulmonary connection indicated a more stable cardiopulmonary condition it is particularly valuable, with possible implications for medical practice.
We thank the reviewer for their positive remark.
Regarding the References, I noticed the presence of only 4 newer citations from 2020, the rest are dominated by the oldest ones, from 1990 to 2010. Even though there are old extremely valuable articles and publications even today, I believe that the authors should expand the list of references with newer publications. This informations can be included in the Discussions chapter and also to do a comparative analysis with the results obtained in the current study.
We have again undertaken a thorough literature review. The available literature on superior cavopulmonary anastomosis in porcine models is limited. However, after an extensive review we found three more citation. These three manuscripts by Kavarana et al published 2013, Sinha et al. published 2018 and Goto et al. published 2023 were included in the „Discussion“ section and discussed (page 20 line 428 to 432 and 452 to 457).
Reviewer 4 Report
Comments and Suggestions for Authors
Dear authours,
Thank you for submitting your manuscript, which presents an interesting and methodologically sound study on the establishment and hemodynamic assessment of superior cavopulmonary anastomosis in a porcine model. Your work provides valuable insights into the acute hemodynamic effects of BDCPC and UDCPC, and the reproducibility of your model is a significant strength. However, I have some concerns about the current data supporting the conclusions:
Major Comments:
- Since the authors declare that this is a reproducible model for future research on partial cavopulmonary connection, more data during the surgery should be provided:
- CPB time
- Arterial blood gas data during the CPB
- Hemodynamic data during the CPB
- The authors noted that the BDCPC group experienced significant cardiopulmonary impairment, it would be helpful to discuss potential mechanisms underlying these changes (e.g., increased ventricular afterload, reduced pulmonary blood flow). Additionally, the authors should discuss the clinical implications of these findings for patients with univentricular physiology.
- This study focuses on acute hemodynamic effects, but it would be valuable to briefly discuss the potential long-term implications of the findings.
Minor Comments:
The authors mentioned that both sexes of animals were used in this study; it is appropriate to briefly discuss whether there were any differences between male and female animals.
Author Response
Reviewer 4:
Thank you for submitting your manuscript, which presents an interesting and methodologically sound study on the establishment and hemodynamic assessment of superior cavopulmonary anastomosis in a porcine model. Your work provides valuable insights into the acute hemodynamic effects of BDCPC and UDCPC, and the reproducibility of your model is a significant strength. However, I have some concerns about the current data supporting the conclusions:
Major Comments:
- Since the authors declare that this is a reproducible model for future research on partial cavopulmonary connection, more data during the surgery should be provided:
- CPB time
- Arterial blood gas data during the CPB
- Hemodynamic data during the CPB
We added requested data on page 19 in an additional table (table 4) and commented these data on page 19, line 393 to 396.
2. The authors noted that the BDCPC group experienced significant cardiopulmonary impairment, it would be helpful to discuss potential mechanisms underlying these changes (e.g., increased ventricular afterload, reduced pulmonary blood flow). Additionally, the authors should discuss the clinical implications of these findings for patients with univentricular physiology.
We appreciate the reviewers remark, however, in the clinical setting in humans a comparable setting to our animal experiments does not exist. We transformed a healthy biventricular heart with competent inflow valves and unobstructed outflow pathways in the BDCPC group into a setting of acute right ventricular afterload increase. Maybe, this scenario is somehow comparable to an acute massive pulmonary embolism. However, in this clinical scenario, insufficient pulmonary blood flow leads to total circulatory collapse. In contrast, in our model, a profoundly impaired circulation persisted due to the patent superior cavopulmonary connection allowing minimal left heart filling via transpulmonary perfusion.
3. This study focuses on acute hemodynamic effects, but it would be valuable to briefly discuss the potential long-term implications of the findings.
Currently, we have only studied the acute hemodynamic changes. No data on the long-term effects are available at the moment. Further research in this regard is planned (page 21, line 486 to 493).
Minor Comments:
The authors mentioned that both sexes of animals were used in this study; it is appropriate to briefly discuss whether there were any differences between male and female animals.
We added on page 3, line 95 the data regarding sex distribution. In both groups two female animals were included each. There was no significant difference in any parameter investigated (page 16, line 333 to 334).
Round 2
Reviewer 1 Report
Comments and Suggestions for Authors
The authors have addressed all the major comments and suggestions made by the reviewer.
The manuscript may be published after editorial check.
Kind regards.
Author Response
We have taken note of the reviewer's favour comment.
Reviewer 4 Report
Comments and Suggestions for Authors
Dear Authors,
I appreciate that you have provided sufficient data addressing my concerns in my previous comments.
Figures 3 and 4 are duplicated.
I would like to recommend that this manuscript be accepted.
Author Response
We thank the reviewer for his accurate review, spotting our paste and copy mistake. Hence we have corrected this mistake and placed Figure 3 and 4 in the correct order.